# A Mathematical Model For Optimal Decisions In A Representative Democracy

**Malik Magdon-Ismail**
Department of Computer Science
Rensselaer Polytechnic Institute
Troy, NY 12180
magdon@cs.rpi.edu

**Lirong Xia**
Department of Computer Science
Rensselaer Polytechnic Institute
Troy, NY 12180
xial@cs.rpi.edu

## Abstract

Direct democracy, where each voter casts one vote, fails when the average voter competence falls below 50%. This happens in noisy settings when voters have limited information. Representative democracy, where voters choose representatives to vote, can be an elixir in both these situations. We introduce a mathematical model for studying representative democracy, in particular understanding the parameters of a representative democracy that gives maximum decision making capability. Our main result states that under general and natural conditions,

1. for fixed voting cost, the optimal number of representatives is *linear*;
2. for polynomial cost, the optimal number of representatives is *logarithmic*.

## 1   Introduction

Suppose a voter-population of size $n$ must vote in a referendum to make an important binary decision to optimize some objective, e.g. social welfare growth over 10 years. A typical solution is *direct democracy* which decides based on a majority vote, the so called "wisdom of the crowd." Direct democracy works when the crowd does indeed have wisdom. In reality, the voters cannot directly observe which decision is correct. Instead, they form beliefs using perceived information, which can be inaccurate, misinterpretable or even manipulated. For example, suppose that each voter's chance to vote for the correct decision, called her *competence*, is i.i.d. generated uniformly over $[0, 0.99]$. Now, the majority among many voters makes the wrong decision with near certainty [17].

This highlights a flaw of direct democracy, where voters participate in decision-making irrespective of their competence. The problems arise in high noise situations, where peoples' beliefs between two choices are nearly split, as in the example above. Such close ties are common in real high-stakes scenarios. For example, in the 2016 United Kingdom European Union membership referendum, 51.89% voted for leave and 48.11% voted for remain. In the 2016 US Presidential Election, 46.1% voted for Trump and 48.2% voted for Clinton.[1] How can democracy cope with high-stakes noisy issues where the average voter competence may drop below 0.5, especially if there is misinformation?

One promising rescue is *representative democracy*, where voters form groups, each group chooses a representative, and the representatives decide via a majority vote. The tradeoff is that there are fewer representatives than base-voters, but, in return, each representative is (hopefully) better informed, being the "wisest" from its group, or at least having a higher competence than the average of its group members. Continuing the example above, let us now use representative democracy, where people are divided into households (e.g. 5 people per group), and let each group choose the member with highest competence as its representative. Then, with high probability the representative's competence is

strictly larger than 0.5. Now, with enough representatives, a majority vote will now make the correct decision with near certainty [17].

It is widely accepted that representative democracy is efficient (lower operational cost and better turnout than direct democracy), yet there is still debate on a fundamental question: *which democracy makes better decisions?* We are not aware of any mathematical framework for analyzing representative democracy w.r.t. its ability to make correct decisions. This is in contrast to direct democracy, which has been mathematically analyzed in depth to provide a justification of the "wisdom of the crowd," which dates back to the *Condorcet Jury Theorem* [13]. Roughly speaking, the Jury Theorem states that a large group of competent voters are likely to make a correct decision by majority voting, which "*lays, among other things, the foundations of the ideology of the democratic regime*" [33]. Direct democracy is just a representative democracy where each group has one voter. Thus, a mathematical characterization of optimal representative democracies would also highlight the subcases where direct democracy is best. The goal of this paper is to establish rigorous mathematical foundations of representative democracy, and provide quantitative answers to the following key questions:

**Q1: What is the optimal number of representatives for representative democracy?**

**Q2: How should electoral group-sizes be distributed (each group has one representative)?**

We will answer these questions in a general setting, where the groups satisfy a weak form of homogeneity with respect to representative election. A concrete example of homogeneity is when each group, which maybe of different sizes, runs the same type of election process on its members who are independent and drawn from some underlying voter-distribution.

In our analysis, we consider two cases. The first is when there is a fixed cost for the voting. In this case, the goal of the representative democracy is to maximize the chances of making the correct decision. This case is relevant to making extremely important decisions where the operational cost of voting is not considered a valid tradeoff for correctness (e.g. presidential election). The second case is when the cost of voting increases with the number of representatives who vote. In this case, one must balance the cost with the benefit that accrues to all $n$ individuals.

**Our Contributions.** We provide a novel mathematical model of representative democracy w.r.t. its ability to make correct decisions and characterize the optimal number of representatives as follows.

1. When the cost of voting is fixed, $\Theta(n)$ representatives is optimal. ($n$ is the population size).
2. When the cost and benefit of voting are both polynomial, $O(\log n)$ representatives is optimal.

In our basic model, there is a single binary issue to be decided and $n$ voters are divided into $L$ groups. Each group chooses a representative by a *representative selection function*, and the representatives will use majority voting to decide a binary issue. Each voter is characterized by her *competence*, which is the probability for her vote to be correct; voter competence is generated i.i.d. from a distribution $F$. Let $\text{Ben}(n) \in \mathbb{R}$ denote the benefit of making the correct decision compared to making the wrong decision for $n$ voters, and let $\text{Cost}(L) \in \mathbb{R}$ denote the operational cost for $L$ representatives to vote.

We reduce representative selection to a *group competence function* $\mu : \mathbb{N} \to [0,1]$, mapping a group's size to the expected competence of its representative. We then extend the Condorcet Jury Theorem to representative democracy by characterizing group competence functions for which the representative democracy makes the correct decision with probability 1 as $n \to \infty$. We informally summarize our main theorems, which are surprising characterizations of the optimal number of representatives.

**Theorems 2, 3, 4 (Optimal representative democracy with fixed voting cost.)** Under natural and mild conditions on the group competence function $\mu$ and when $\text{Cost}(L) = \text{constant}$:

1. (Homogeneous groups) The optimal group size $K^*(n)$ is at most a constant independent of $n$.
2. (Inhomogeneous groups) The optimal number of representatives $L^*$ is linear in $n$.
3. The optimal group size distribution is nearly homogeneous.

These results hold, independently of the specific details of the representative selection process. Let us highlight why the result is unexpected in a concrete context. Suppose voter competence is drawn from some continuous density on $[0, 1]$ and a group elects the most competent of its voters as representative (very optimistic since it cannot get better than that). Then by choosing larger and larger groups, the competence of the representative approaches 1. The price paid is that there are fewer of these ultra-smart representatives, but there will still be many of them as $n \to \infty$. Indeed, one might posit that some optimal tradeoff exists whereby the group size tends to infinity but at a slower rate than $n$, so that the representatives get smarter and smarter *and* there are more and more representatives. Our theorem establishes the contrary. The optimal group size will never exceed some constant (the constant's value may depend on the specific parameters of the selection process).

To prove our results, we use novel combinations of combinatorial bounds. In addition, some of our results may be of independent interest, for example, Lemma 7 in the full version answers an open question on the probability for majority voting to be correct when the average competence of voters is exactly $0.5$ where there are no general results for the non-asymptotic behavior [17] and the asymptotic behavior is only conjectured [32, Lemma 5].

We then consider the case of polynomial cost and polynomial benefit: for $q_1, q_2 > 0$, $\mathrm{Cost}(L) = L^{q_1}$ and $\mathrm{Ben}(n) = n^{q_2}$. A special case is linear cost and linear gain, where $q_1 = q_2 = 1$. The cost of voting has a significant impact on the optimal group size.

**Theorem 5 (Optimal representative democracy with polynomial cost and benefit).** Under natural and mild conditions on the group competence function $\mu$, when $\frac{\mathrm{Cost}(L)}{\mathrm{Ben}(n)} = \Theta(\frac{L^{q_1}}{n^{q_2}})$:

1. The optimal number of representatives is in $O(\log n)$.
2. When $\mu(K)$ polynomially converges to $1$, the optimal number of representatives is in $\Theta(\log n)$.
3. If $\mu(K)$ is bounded below $1$, the optimal number of representatives is in $\Theta(1)$.

Our analysis can be extended to the situation where representatives will have to vote on $d > 1$ possibly correlated issues. This is the case with the US Senate and House members, who after election may have to cast a vote on multiple occasions on different issues (about once per week). We defer these results to the full version.

**Related Work and Discussion.** In the US, each House member represents about 700K people, which used to be as low as 33K. Many favor enlarging the House [18, 3, 22]. We provide a mathematical foundation to analyze such choices. Mathematical models of representative democracy exist, e.g. [10, 1], however none quantitatively characterizes optimal representative democracy w.r.t. its decision-accuracy. Our work is related to extensions of The Condorcet Jury Theorem to heterogeneous agents [29, 21, 30]. Along this vein are three subareas: (1) understanding the conditions for consistency of the majority [32, 33, 24, 17, 37], (2) studying optimal population size to maximize correctness [16, 27, 20, 34, 7, 4, 28, 26, 8, 9, 38, 6], and (3) incentivizing voters to increase their competence [29, 25, 5]. See [31] for a recent survey, including extensions to strategic voters.

Our work is related to the first two subareas (consistency and optimal size). The key differences are, first, in our work, the competence of the representatives is endogenous to our model, a result of partitioning and representative selection, while the competence of voters in the literature is given. Second, in our work, our setting has a tradeoff: increasing the number of representatives means weaker representatives, and hence may decrease the overall correctness. We give quantitative analysis of the quality vs. quantity tradeoff in representative democracy. Our results also extend to multiple issues (discussed in the full version).

Our work is related to some recent work in proxy voting or liquid (delegative) democracy [12, 23]. Our voting dynamic is different because voters are not allowed to delegate to an arbitrary voter, and representatives cannot delegate their votes. Technically, we require minimal assumptions on the representative selection process. We also consider cost of voting, which is not considered in [12, 23]. There is recent work in computational social choice on using statistics to make better decisions [14, 11, 40, 41, 36, 35, 15, 2, 39], which focuses on direct (not representative) democracy.

## 2    Mathematical Model of Representative Democracy

In this section we propose a mathematical model for representative democracy for one issue. As in the Condorcet Jury Theorem, we assume that voters' ability to vote for the correct decision is drawn i.i.d. from a distribution $F$. The voters are divided into $L \geq 1$ groups. For any group $\ell$ with $K$ voters, whose competences are $\{q_{\ell,1}, \ldots, q_{\ell,K}\}$, a deterministic or randomized representative selection process $V_\ell$ chooses a member $q_\ell$. The chosen representative casts a vote, and majority voting succeeds if strictly more than half of representatives vote $1$. The process for a group to choose a representative to vote is summarized below.

$$ F \xrightarrow{\text{generate voters}} \{q_{\ell,1}, \ldots, q_{\ell,K}\} \xrightarrow{\text{elect representative}} q_\ell \xrightarrow{\text{cast vote}} x_\ell $$

**Definition 1.** *A representative democracy for one issue is composed of the following components.*

- **Issue.** *Suppose there is one issue to decide, whose outcome is $1$ (correct), and $0$ (incorrect).*

- **Partition function** $\vec{K}$**.** *For any number of voters* $n$, $\vec{K}$ *denotes a partition function that divides* $n$ *voters into* $L(n)$ *groups, that is,* $\vec{K}(n) = (K_1, \ldots, K_{L(n)}) > 0$ *and* $\sum_{i=1}^{L(n)} K_i = n$.

- **Distribution of competence** $F$**.** *We assume that each voter's* type *is characterized by her competence, which is the probability for her to vote for* 1*. Each voter's competence is i.i.d. from a distribution* $F$ *over* $[0, 1]$.

- **Representative selection process** $V_\ell$**.** *Suppose group* $\ell$ *has* $K$ *users whose competences are* $q_{\ell,1}, \ldots, q_{\ell,K}$ *respectively. The group uses a (randomized) process* $V_\ell(q_{\ell,1}, \ldots, q_{\ell,K})$ *to select a representative* $q_\ell$, *whose vote is represented by a random variable* $x_\ell$.

- **Voting by the representatives.** *The chosen representatives will vote for* 1 *with probability equivalent to her competence, and vote for* 0 *otherwise. The majority rule is used to decide the outcome based on representatives' votes. We assume that a strict majority of votes is necessary to make the correct decision. That is, when there is a tie, the outcome is* 0 *(incorrect).*

When each group has exactly 1 member, we obtain the direct democracy, otherwise we have the representative democracy as in the following example.

**Example 1** (Uniform Voters with Uniform Process)**.** Suppose there are $L \geq 1$ groups, each group has $K \geq 1$ members. Each voter's competence is i.i.d. from Uniform$[a, b]$. For all groups, $V_\ell$ chooses a member uniformly at random. It is not hard to see that majority voting by the representatives is correct with probability no more than $0.5$ when $a + b < 1$. ∎

We note that the vote of group $\ell$'s representative, i.e. the random variable $x_\ell$, is characterized entirely by its expectation, which contains all information needed in the analysis in this paper. Therefore, we will simplify the representative selection process to a single *group competence function* $\mu$, which specifies the expected competence of the representative as a function of the group size $K$.

**Definition 2.** *A representative democracy is a* group competence function $\mu : \mathbb{N} \mapsto [0, 1]$.

For example, the group competence function for the uniform process in Example 1 can be represented by $\mu_{\mathrm{U}}(K) = (a + b)/2$ for all $K$. We will see that the group competence function significantly simplifies the process in the following two examples. In the next example, the group chooses its most-informed member—the one with the highest competence—as the representative.

**Example 2** (Max Process)**.** As in Example 1, suppose $F = \text{Uniform}[a, b]$. Each group now chooses the member with the highest competence as the representative. So $q_\ell = \max\{q_{\ell,1}, \ldots, q_{\ell,K}\}$, which means $(q_\ell - a)/(b - a) \sim \text{Beta}(K, 1)$ [19], from which $\mu_{\max} = \mathbb{E}[x_\ell] = (a + Kb)/(K + 1)$. ∎

The group competence function for MAX process is monotonically increasing in $K$, between $a$ and $b$, while the group competence function for UNIFORM process outputs the same value for all $K$. The MAX process is an upper bound on group competence functions. Competence may not be observable as in Example 2. However, when the group size is not too large, for example a group is a household, then it is natural to assume that the family members are able to choose the max-informed representative. Even if the process is noisy, it is reasonable to expect more competent voters to have higher probability to become the representative, as in the noisy-max process below.

**Example 3** (Noisy-Max Process)**.** Continuing with Example 1, let $\vec{p} = (p_1, \ldots, p_K)$ satisfy $p_1 \cdots \geq p_K \geq 0$ and $\sum_{i=1}^{K} p_i = 1$. For $i \leq K$, choose the member with $i$-th highest competence as the representative with probability $p_i$. Then, $\mu_{\vec{p}}(K) = (\vec{p} \cdot \vec{k}_+)a + (\vec{p} \cdot \vec{k}_-)b$, where $\vec{k}_+ = \frac{1}{K+1}(1, 2, \ldots, K)$ and $\vec{k}_- = \frac{1}{K+1}(K, K-1, \ldots, 1)$. ∎

## 3 Optimal Representative Democracy for One Issue

We first extend the classical Condorcet Jury Theorem to representative democracy. Let us formally define consistency, the main desired property of a democracy. As the population increases, i.e. asymptotic in $n$, it should be possible to partition the voters into $L$ groups, with potentially different number of voters in each group, such that with probability 1 the majority representatives vote for 1.

Given $\vec{K} = (K_1, \ldots, K_{L(n)})$, we let $S_{n, \vec{K}, \mu}$ be the fraction of 1's in $L(n)$ independent Bernoulli random variables with success probabilities $(\mu([\vec{K}(n)]_1), \ldots, \mu([\vec{K}(n)]_{L(n)}))$, where for $i \leq L(n)$, $[\vec{K}(n)]_i = K_i$ is the $i$-th component of $\vec{K}(n)$.

**Definition 3.** *Given a partition function $\vec{K}$ and a group competence function $\mu$, for any $n$, we let $R_n(\vec{K}, \mu)$ denote the probability for majority voting by representatives according to $\vec{K}$ and $\mu$ to be correct. That is, $R_n(\vec{K}, \mu) = \mathbb{P}(S_{n,\vec{K},\mu} > \frac{1}{2})$. (We use $R(\vec{K}, \mu)$ when $n$ is clear from the context.)*

**Definition 4.** *A representative democracy with group competence function $\mu$ is* consistent *if there exists a partition function $\vec{K}(n)$ for which the majority voting of representatives is correct with probability 1 as $n \to \infty$, that is, $\lim_{n \to \infty} R_n(\vec{K}, \mu) = 1$.*

**Theorem 1.** *For a voter-competence distribution $F$, the representative democracy with group competence function $\mu$ is consistent if and only if there exists $K_* \in \mathbb{N}$ such that $\mu(K_*) > 0.5$.*

Missing proofs are in the full version. We now define the benefit, cost, and social welfare.

**Definition 5.** *Let $Ben(n) \in \mathbb{R}$ be the benefit of making the correct decision and let $Cost(L)$ be cost of maintaining $L$ representatives. Given a partition $\vec{K} = (K_1, \ldots, K_{L(n)}) \in \mathbb{N}^{L(n)}$, the* social welfare *of $\vec{K}$ is the expected benefit minus the cost of voting, that is,*

$$SW(\vec{K}) = Ben(n)R_n(\vec{K}, \mu) - Cost(L) = Ben(n)\left(R_n(\vec{K}, \mu) - \frac{Cost(L)}{Ben(n)}\right)$$

## 3.1 Optimal Group Size for Fixed Cost of Voting

In this section, we focus on a fixed voting cost regardless of the number of representatives. In this case, the goal is to find the optimal partition $\vec{K}$ that maximizes $R_n(\vec{K}, \mu)$. Let us first informally discuss the effect of increasing the size of groups, which trades quality (competence of each representative) for quantity (the number of representatives), and is a form of mean vs. variance tradeoff.

**The quality vs. quantity tradeoff.** Suppose $n$ is fixed and we are deciding between group sizes $K_1 < K_2$, with $\mu(K_2) \geq \mu(K_1) > 0.5$. For simplicity, suppose $K_1$ and $K_2$ divide $n$ and assume group sizes are equal in both cases, so $\vec{K}_1 = (K_1 \ldots, K_1)$ and $\vec{K}_2 = (K_2 \ldots, K_2)$. With $K_1$ members in each group, each representative has competence $\mu(K_1) < \mu(K_2)$. On the other hand, the number of representatives is $L_1 = n/K_1 > n/K_2 = L_2$. Therefore, we have $\mathbb{E}(S_{n,\vec{K}_1,\mu}) = \mu(K_1) \leq \mu(K_2) = \mathbb{E}(S_{n,\vec{K}_2,\mu})$, while the variance of $S_{n,\vec{K}_2,\mu}$, which is $\mu(K_1)(1-\mu(K_1))/L_1$, can potentially be smaller than the variance of $S_{n,\vec{K}_2,\mu}^2$.

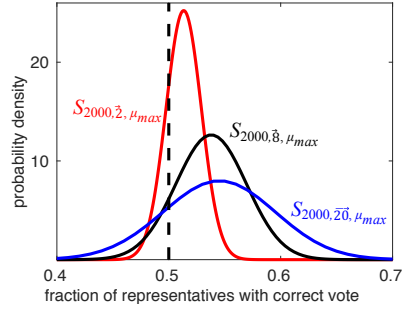

Figure 1: The mean vs. variance trade-off when selecting group size.

Therefore, the optimal group size $K^*$ minimizes the left tail probability $\mathbb{P}[S < 0.5]$. This mean vs. variance tradeoff is illustrated using the distribution of $S$ in Figure 1 for the MAX process with $n = 2000$ and $F = \text{Uniform}[0.44, 0.55]$. We compare three group sizes: $K = 2$ (low mean, low variance), $K^* = 8$ (optimal, middle), and $K = 20$ (high mean, high variance).

Naturally, our next goal is to identify an optimal number of representatives (groups), denoted by $L^*(n)$. We will first focus on the specific partition functions where almost all groups have the same size, with the exception for the last group, whose size is allowed to be larger than other groups. This is a natural setting in practice because each representative is supposed to represent equal number of voters. Later in this section we will show how to extend our study to general partition functions.

**Definition 6** (Homogeneous groups). *Given $n$ and a group size $K$, in the homogeneous setting, $L = \lfloor n/K \rfloor$ representatives are selected using partition function $\vec{H}_K = (K, \ldots, K, n-(L-1)K)$, with $L - 1$ groups of size $K$.*

Our main result is that for homogeneous groups, the optimal group size is bounded by a constant, provided that some value of $K$ can achieve consistency and the group competence function $\mu$ is polynomially bounded away from 1 as $K$ goes to infinity. Let $K_{\text{hom}}^*(n)$ denote the optimal homogeneous group size that maximizes $SW(\vec{H}_K)$, where $Cost(L)$ is a constant. Then, $K_{\text{hom}}^*(n)$ maximizes $R_n(\vec{H}_{K_{\text{hom}}^*(n)}, \mu)$, the probability for the majority of representatives to be correct.

**Theorem 2** (Optimal homogeneous group size). *Suppose the $Cost(L)$ is a constant, there exists $K_* \in \mathbb{N}$ such that $\mu(K_*) = \frac{1}{2} + \epsilon$ for $0 < \epsilon < \frac{1}{2}$, and for all $K \in \mathbb{N}$, $\mu(K) \leq 1 - A/K^\alpha$ for*

*constants $A > 0$ (w.l.o.g. $A \leq 1$) and $\alpha \geq 0$. Then, $K_{hom}^*(n) \leq cK_*$, where*

$$c = \frac{-4}{\ln(1 - 4\epsilon^2)} \left( \ln \frac{32}{9\epsilon^2 A} + \alpha \ln \frac{4\alpha K_*}{e} - \alpha \ln |\ln(1 - 4\epsilon^2)| \right).$$

The intuition is that as $K$ increases, we experience diminishing returns with respect to $\mu$ because $\mu$ is bounded away from 1. On the other hand, there is a loss due to decreasing $L = \lfloor n/K \rfloor$, the number of representatives. One may expect that the best tradeoff is achieved when $K$ is a slowly increasing function of $n$, but our theorem proves otherwise. The proof is technical and involves a subtle analysis of the tail probabilty $\mathbb{P}[S < 0.5]$, which is deferred to the appendix.

*Proof.* (Sketch) We may assume the first $L-1$ groups has $K$ members. We first observe an elementary bound that allows us to ignore the last group whose number of representatives is unknown:

$$\underbrace{\mathbb{P}\left[ \sum_{\ell=1}^{L-1} x_\ell \geq \left\lceil \frac{L+1}{2} \right\rceil \right]}_{R_-(L,\mu(K))} \leq R_n(\vec{H}_K, \mu) \leq \underbrace{\mathbb{P}\left[ \sum_{\ell=1}^{L-1} x_\ell \geq \left\lceil \frac{L-1}{2} \right\rceil \right]}_{R_+(L,\mu(K))}.$$

The functions $R_-$ and $R_+$ are Binomial upper tail probabilities. We have not yet invoked any properties of the group competence function $\mu$. The following lemmas are properties of $R_\pm(L, p)$.

**Lemma 1** (Monotonicity of $R(L, p)$). *For fixed $L$, $R_-(L, p)$ and $R_+(L, p)$ are increasing in $p$. For fixed $p > \frac{1}{2}$, $R_-(L, p)$ and $R_+(L, p)$ are increasing in $L$.*

**Lemma 2.** *For $p \leq \frac{1}{2}$, $R_+(L, p) \leq \frac{3}{4}$ (the maximum is attained for $L = 3, p = \frac{1}{2}$).*

**Lemma 3** (Binomial tail inequality). *Given $p > \frac{1}{2}$, $L$ and $k \leq \lceil L/2 \rceil$,*

$$\binom{L}{k} p^k (1-p)^{L-k} \leq \sum_{\ell=0}^{k} \binom{L}{\ell} p^\ell (1-p)^{L-\ell} \leq \frac{p}{2p-1} \binom{L}{k} p^k (1-p)^{L-k}.$$

**Lemma 4** (Near-Central binomial coefficient bound). *For $L > 1$,*

$$\frac{3}{4} \cdot \frac{4^{L/2}}{\sqrt{\pi L}} \leq \binom{L}{\lceil \frac{1}{2}(L-1) \rceil} \leq 2 \cdot \frac{4^{L/2}}{\sqrt{\pi L}}.$$

**Lemma 5** (Bounding $R_-(L, p)$ and $R_+(L, p)$). *For $\frac{1}{2} < p < 1$,*

$$1 - \left( \frac{2}{(2p-1)} \right) \cdot \frac{(4p(1-p))^{L/2}}{\sqrt{\pi L}} \leq R_-(L, p) \leq R_+(L, p) \leq 1 - \frac{3}{8p} \cdot \frac{(4p(1-p))^{L/2}}{\sqrt{\pi L}}.$$

We are ready to prove the theorem. Let $c$ be defined as in the statement of the theorem. We may assume $n > cK_*$ otherwise the theorem automatically holds, because $K_{hom}^*(n) \leq n \leq cK_*$. Further, if $L_* = 1$, then there is just one group and any $K > K_*$ will also have just one group and be equivalent. Therefore, we may assume $L_* \geq 2$. Now suppose $K > cK_*$. Define $\mu_K = \mu(K)$ and $L_K = \lfloor n/K \rfloor$, $\mu_* = \mu(K_*)$ and $L_* = \lfloor n/K_* \rfloor$. Observe that $L_K \leq L_*$. We show that $R(L_*, \mu_*) \geq R(L_K, \mu_K)$ which means that $K$ cannot be better than $K_*$ for a homogeneous partition of $n$, proving the theorem.

If $\mu_K \leq \frac{1}{2}$ then $R_+(L_K, \mu_K) \leq \frac{3}{4}$ (Lemma 2). We show that $R_-(L_*, \mu_*) > \frac{3}{4}$. Indeed, since $n > cK_*$, we have $n/K_* > c \geq \frac{-4}{\ln(1-4\epsilon^2)} \cdot \ln \frac{32}{9\epsilon^2 A}$, and so

$$L_* = \left\lfloor \frac{n}{K_*} \right\rfloor \geq \frac{n}{2K_*} > \frac{-2}{\ln(1 - 4\epsilon^2)} \cdot \ln \frac{32}{9\epsilon^2 A} \qquad \Longrightarrow \qquad \frac{(1-4\epsilon^2)^{L_*/2}}{\epsilon\sqrt{\pi L_*}} < \frac{9\epsilon A}{32\sqrt{\pi L_*}} < \frac{1}{4},$$

where in the last inequality we used $A \leq 1$ and $L_* \geq 2$. Using the bound for $R_-$ from Lemma 5 gives $R_-(L_*, \mu_*) > \frac{3}{4}$, which proves $K$ is not optimal. Therefore, we may assume that $\mu_K > \frac{1}{2}$. Also, if $\epsilon = \frac{1}{2}$ then the representatives always vote for the correct decision and the theorem automatically holds, so we may assume $\epsilon < \frac{1}{2}$. Now, using Lemma 5, $L_* \geq L_K$ and some algebra:

$$
\begin{aligned}
R_n(\vec{H}_{K_*}, \mu) - R_n(\vec{H}_K, \mu) &\geq R_-(L_*, \mu_*) - R_+(L_K, \mu_K)] \\
&\geq (\text{positive}) \cdot \left[ 1 - C \left( \frac{(4\mu_*(1-\mu_*))^{L_*/L_K}}{4\mu_K(1-\mu_K)} \right)^{L_K/2} \right],
\end{aligned}
$$

where $C = \frac{16}{3} \cdot \frac{\mu_K}{2\mu_* - 1} = \frac{8\mu_K}{3\epsilon}$. Note that $C \leq \frac{8}{3\epsilon}$ (because $\mu_K < 1$) and $C > 1$ (because $\mu_K > \frac{1}{2}$ and $\epsilon < \frac{1}{2}$). We prove the term in square parentheses is positive. Observe that

$$\frac{L_*}{L_K} = \frac{\lfloor n/K_* \rfloor}{\lfloor n/K \rfloor} \geq \frac{\lfloor n/K_* \rfloor}{n/K} = \frac{K}{K_*} \frac{\lfloor n/K_* \rfloor}{n/K_*} \geq \frac{K}{2K_*}.$$

We used $\lfloor x \rfloor / x \geq \frac{1}{2}$ when $x \geq 1$. Because $\mu_K > \frac{1}{2}$ and $1 - \mu_K \geq A/K^\alpha$, we have $\mu_K(1 - \mu_K) \geq A/2K^\alpha$. Also, $\mu_* \geq \frac{1}{2} + \epsilon$, which means that $\mu_*(1 - \mu_*) \leq (\frac{1}{2} + \epsilon)(\frac{1}{2} - \epsilon)$, and $K \leq n$. Therefore,

$$C \left( \frac{(4\mu_*(1 - \mu_*))^{L_*/L_K}}{4\mu_K(1 - \mu_K)} \right)^{L_K/2} \leq C \left( \frac{K^\alpha(1 - 4\epsilon^2)^{K/2K_*}}{2A} \right)^{L_K/2}$$

We show that the RHS is at most 1, or equivalently its logarithm is at most zero, concluding the proof. Taking the logarithm of the RHS, we get:

$$L_K \left( \frac{K}{4K_*} \ln(1 - 4\epsilon^2) + \frac{\alpha}{2} \ln K - \frac{1}{2} \ln 2A \right) + \ln C$$

$$\overset{(L_K \geq 1, C > 1)}{\leq} L_K \left( \frac{K}{4K_*} \ln(1 - 4\epsilon^2) + \frac{\alpha}{2} \ln K - \frac{1}{2} \ln 2A + \ln C \right)$$

$$\leq L_K \left( \frac{K}{4K_*} \ln(1 - 4\epsilon^2) + \frac{\alpha}{2} \left( \ln \frac{-4\alpha K_*/e}{\ln(1 - 4\epsilon^2)} - \frac{\ln(1 - 4\epsilon^2)}{4\alpha K_*} K \right) - \frac{1}{2} \ln 2A + \ln \frac{8}{3\epsilon} \right).$$

The last step follows by using the fact for any $z > 0$, $\ln x \leq \ln(z/e) + x/z$, which holds because for any $y = x/z > 0$, $\ln y - y + 1$ is maximized at $y = 1$. In the last step, we set $z = -4\alpha K_*/\ln(1 - 4\epsilon^2)$. To conclude the proof, collecting terms and use $K/K_* > c$ and $\ln(1 - 4\epsilon^2) < 0$. ∎

As a corollary, Theorem 2 can be applied to any $F$ with continuous density function, which means that in such cases the optimal number of representatives with homogeneous group-size is $\Omega(n)$.

**Corollary 1** (Linear number of representatives). *Suppose $F$ has a continuous density function on $[0, 1]$. For any $\mu$ such that $\mu(K_*) > 0.5$ for some $K_*$. The optimal number of representatives for homogeneous groups is at least $\frac{n}{cK_*}$, where $c$ is the constant defined in Theorem 2.*

When costs are fixed, a representative democracy with fixed group size makes better choices than the representative democracy with fixed number of representatives. One limitation of Theorem 2 is that it only holds for homogeneous group size. We can drop this restriction if the group competence function $\mu(K)$ is concave (for example the MAX process is concave). Let $L^*(n)$ denote the optimal number of groups for $n$ voters.

**Theorem 3** (Optimal number of representatives for general group sizes). *Suppose $Cost(L)$ is a constant, there exists $K_* \in \mathbb{N}$ such that $\mu(K_*) \geq \frac{1}{2} + \epsilon$ for $0 < \epsilon < \frac{1}{2}$, $\mu$ is concave, and $\mu(K) \leq 1 - A/K^\alpha$ for constant $A < 1$ and $\alpha \geq 0$. Then, for any $n$, $L^*(n) \geq \lfloor \frac{n}{K_*} \rfloor / c$, where*

$$c = \frac{-4}{\ln(1 - 4\epsilon^2)} \left( \ln \frac{32}{9\epsilon^2 A} + \alpha \ln \frac{4\alpha K_*}{e} - \alpha \ln |\ln(1 - 4\epsilon^2)| \right).$$

The proof of Theorem 3 is similar to the proof of Theorem 2. Next, we extend Theorem 3 by showing that the optimal partitioning function $\vec{K}$ is nearly homogeneous if $\mu$ is log-concave and $1 - \mu$ is log-convex. Log-concavity is weaker than concavity (e.g., the MAX process is log-concave).

**Theorem 4** (Near-Homogeneity of group sizes). *Suppose $Cost(L)$ is a constant, the group competence function $\mu(K)$ is log-concave and non-decreasing, and further that $1 - \mu(K)$ is log-convex. Given $L$ groups, there is an optimal partition $(K_1, \ldots, K_L)$ of $n$ into the $L$ groups with no two groups differing in size by more than 1. That is, $\max_i K_i - \min_i K_i \leq 1$.*

Theorem 4 also applies to non-constant cost functions because the number of groups is fixed.

**Numerical Example of Optimal Group Size.** As an application of our results, we consider the uniform voters with MAX representative selection process (Example 2) within a simple setting which capturea at a very coarse level the US House of Representatives. We pick a voter distribution $F = \text{Uniform}[0.45, b]$ (of course, this may not capture voters in the US, but it is just illustrative.) Below, we show how the optimal homogeneous group size $K^*$ and the minimum group size required to achieve consistency $K_*$ depend on $b$. When $b$ is small, as voters get wiser ($b$ goes up), $K_*$ and $K^*$ are decreasing. At some point, even the direct-democracy ($K_* = 1$) is consistent. For very large $b$, the optimal group size starts to increase due to the $\mu(1-\mu)$ term. In general, the optimal

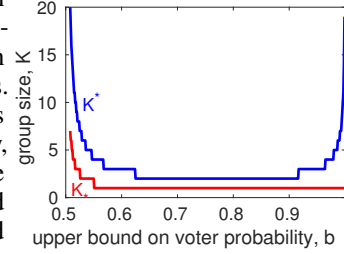

Figure 2: Optimal group size.

group size is less than about 5, the size of the typical household. The optimal representative democracy is obtained when each household elects a head to vote on its behalf.

The optimal group size is useful to know, but for practical purposes, there may not be a significant difference between different values of $K$ for a large $n$. Let us take the US House as an example, which has 435 representatives. Suppose the voting population is about $n = 235$ million, and that voter competence is uniformly distributed from 0.45 to 0.52 (the average competence is slightly less than 0.5), which corresponds to $b = 0.52$ in our example.

|  | $K = 1$ | $K_* = 3$ | $K^* = 9$ | $5 \times$ House | $2 \times$ House | House |
|---|---|---|---|---|---|---|
| Success rate | 0% | 100% | 100% | 97% | 88% | 80% |

In this simple setting, direct democracy ($K = 1$) would be wrong, and the current size of the House is far from optimal, 20% less accurate than what is achievable. Doubling congress gets you to 88% and multiplying by 5 pretty much gets you to optimal. A House that is 20 times larger (each member representing 35K citizen) would be essentially indistinguishable from optimal. This suggests that a much larger House is needed for noisy issues like the one in this example. Also note that if group sizes are about 5 (the size of a household) then we have near-perfect results.

## 3.2 Optimal Group Size for Polynomial Cost of Voting

When there is a non-constant cost to voting, things change dramatically. We consider the case where the cost and benefit are both polynomial here, though our analysis can be extend to other functional forms, with different results. Perhaps the most intuitive cost and benefit are both linear (per-representative cost and per-voter benefit). We simply state the result.

**Theorem 5** (Optimal homogeneous group size, polynomial cost and polynomial benefit)**.** *Suppose* $\frac{Cost(L)}{Ben(n)} = \Theta(\frac{L^{q_1}}{n^{q_2}})$ *for constants* $q_1 > 0$ *and* $q_2 > 0$*, there exists* $K_* \in \mathbb{N}$ *such that* $\mu(K_*) > \frac{1}{2}$*,* $\mu$ *is non-decreasing, and for all* $K \in \mathbb{N}$*,* $\mu(K) \leq 1 - A/K^\alpha$ *for constants* $A > 0$ *and* $\alpha \geq 0$*. Then, the optimal group size* $K^*_{hom}(n) = \Omega(n/\log n)$*. Moreover, we have:*

(i) *If* $\lim_{K \to \infty} \mu(K) < 1$*, then* $K^*_{hom}(n) = \Theta(n/\log n)$*.*

(ii) *If there exists* $B, \beta > 0$ *such that* $\mu(K) \geq 1 - B/K^\beta$*, then* $K^*_{hom}(n) = \Theta(n)$*.*

## 4 Summary and Future Work

We set the mathematical foundation for studying the optimal number of representatives in a representative democracy and showed that under general and natural conditions, the optimal is linear when the cost of voting is a constant, and logarithmic when the cost and benefit are both polynomial. Our results can be extended to multiple issues. There are many open questions: Can we extend to inhomogeneous representative selection processes, e.g. different states use different processes to choose representatives? Does diversity (inhomogeneous agents) in population help make better decisions? What happens if agents are strategic?

## Acknowledgements

We thank all anonymous reviewers for helpful comments and suggestions. LX is supported by NSF #1453542 and ONR #N00014-17-1-2621.

## Footnotes

[1] These examples are only used to show real situations where close ties exist. We do not know if direct democracy would succeed or fail in these cases since we do not know what the "correct" outcome is.

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
