[Reviews · NeurIPS 2018]

Reviewer 1



The paper models representative democracy for n voters by dividing voters into L groups (often homogenous with K = n/L + O(1) members), having each group pick a representative, and choosing the issue at hand via majority vote of the representatives. The representative choice mechanism is imagined to pick an "above average" voter in terms of voting skill, but only weak assumptions on this mechanisms are required for the main theorems. If the cost of voting is constant, the main theorem is that optimal group size is O(1): more accurate decisions are made by many small groups, and the ideal group size is bounded as the number of voters n -> infinity. If the cost of voting and the benefit of a correct decision are both polynomial (in L and n, respectively), the ideal number of representatives is O(log n), with more precision with extra assumptions. This is a nonstandard paper for NIPS, but it tackles interesting questions and the results are nonobvious. I would argue for acceptance, and would argue more strongly if concerns about field match are ignored. It is possible there are c onnections between these results and ensembling techniques, or in using majority vote or other schemes to improve data quality when gathering data from humans. My main high level comment is that the proofs are all effective or very close to effective, and in particular Theorem 2 computes the complicated exact constant c. The proofs could be simplified if this effectiveness was dropped, and it's unclear what the benefit of knowing the form of the constant is since I am not confident that it's tight (there is no discussion of this). I'll leave this decision up to the authors. Comments: 1. I'm not sure what the imagined final paper structure is, since the supplementary material is just a longer version of the whole paper rather than structured as appendices. 2. Line 98 has "Cost(n) = n^q2", but should be "Ben(n)". 3. In the proof of lemma 3, you say "at least a factor µ/(1 − µ) bigger than the previous term". This should be p/(1-p). 4. Lemma 5's proof is full of unnecessary parentheses around 2p-1. 5. In the proof of lemma 5, worth a note that one of the inequalities is because p/(1-p) > 1. This is pretty subtle since you drop a floor without comment, and p/(1-p) > 1 is needed to make that safe. 6. Typo: "resteiction" (there are likely others). 7. In the proof of theorem 5, "p(L)" is unfortunate notation given that p = mu is used elsewhere.

Reviewer 2



The paper studies group-based majority voting. There is a binary issue with a "correct" answer. Each voter has some probability of voting the right way. In the proposed set up, voters are organized into groups of K, and the final decision is a majority vote among the highest-probability voter in each group. (This is the "max" aggregation function; the authors also study some other options.) The main finding are that with constant voting cost the optimal group size is constant, while with polynomial voting costs it is near-linear. The model is reasonably natural and topical. On a technical level, the arguments are elementary probability calculations --- not trivial, but not particularly illuminating, either.

Reviewer 3



Summary: - In this paper, the authors have presented a framework of mathematically exploring the trade-offs of representational democracy, when the decisions are taking in a hierarchical fashion: first a group of some size K selects a representative from among them using some group selection function \mu and then the representatives cast their votes on the issue at hand and the majority wins. - Their framework assumes that each voter's competence is IID and a common assumption for most "interesting" results presented in the paper is that the group selection function \mu is such that there exists a group size for which the expected vote for the 'correct' side can be made > 1/2. - The framework allows modeling the benefit of selecting the correct decision (which depends on the size of the population) and the cost of casting a vote (which depends on the size of the set of representatives) and different distributions for the competence. - In the proofs, the authors first establish bounds for the "homogeneous" setting, where the size of all representative groups, save the last, are the same size and then show that the optimal group size would be "nearly" homogeneous: i.e. would differ by at most one (with some additional assumptions about the representative selection function). - There are some surprising findings in the paper: * if we ignore the cost of casting votes, then the optimal group size is a constant which depends on the group selection function and the underlying distribution of competence, but which is independent of the size of the population. This means that the number of representatives is linear in the size of the population. * if the cost of casting a vote and the benefits are polynomial, then the optimal group size grows roughly as O(n / log n). This means that the number of representatives is logarighmic in the size of the population. - The authors also generalize their framework to consider representatives which have to cast votes on multiple issues. - Along the path, they answer an open question about what is the probability of attaining a majority when "competence" of voters is exactly 0.5 and show, somewhat surprisingly, that the probability of majority depends heavily on whether the number of voters is odd or even. ---- This is an interesting paper which tackles an important problem. It proposes novel mathematical framework for analysis of optimal decisions in Representative Democracy. The paper is a "first-principles" approach to the problem and presents several interesting results which are begging for more exploration. The presentation of the paper is logical. The examples and the intuitive explanations help in understanding the Theorems. However, such explanations dry up as we advance through the paper. For example, making the intuition behind polynomial costs of voting is found wanting. The results themselves are interesting and look worth exploring further. They involve innovative usage of combinatorial and probabilistic inequalities. However, I believe that the presentation of the proofs paper can be made simpler for the reader to follow. In particular, abuse of notation in the paper is rampant, which makes the proofs very difficult for me to follow. A few helpful hints to improve the understanding of first time readers: - Is \mu a function from a set to a number or from a natural number to a number? I understand that because of the IID assumption, it could be stated as either, but defining it as one and using it as the other appears sloppy and causes confusion. - Why does n < cK_{*} cause Theorem 2 to hold automatically? Is there some property of \mu makes it such that selecting just one group will be optimal? The reason is probably simple, but I had difficulty in establishing it. - Why is c > (-4/(ln (1 - 4eps^2)) * ln(32/9Aeps^2))? I was not obvious to me that the remaining part of the constant 'c' was negative in value. I do not think that these issues will end up invalidating the proof: I could see that the theorems would likely be true even with corrections (if any). Hence, my support for the paper. The supplementary material contains interesting results for multi-issue voting, which are not mentioned at all in the main body? Could the proofs be relegated to the appendix and the statement of the results for multiple issues moved to the main text? The theoretical bounds in the paper seems ripe for simulations which would open another window in the behavior of the algorithms in different settings: e.g. what kind of deviations may one see while moving away from the optimum group sizes/number of representatives. In contrast, the experimental results presented, though interesting and relevent, seem a bit sparse. Finally, in the abstract, the authors pose this as a problem of creating ensembles of classifiers. However, beyond that, there are no references to machine learning methods. Looking at the list of references too suggests that this paper would perhaps be a better fit for an avenue which focusses on mathematical analysis of social choices. Though I believe that the paper explores an important problem and presents interesting results and is likely to be of some general interest, due to the caveats with a potentially mismatched scope, somewhat weak empirical analysis, and some issues with the presentation decisions, I would overall recommend rejection. Minor issues: - line 228: "capturea" -> "captures" - Example 5 (supplementary): What does (1/2)@c1 mean? - Theorem 2: Why is A <= 1 w.l.o.g.? If A > 1 => the inequality \mu(K) <= 1 - A/K^a will never be satisfied for K = 1. Hence, A \in (0, 1] is a requirement. ---- Update: Reading the other reviews, I am convinced that the paper makes an interesting contribution. Reading the response of the authors, I am convinced that the final version of the paper will address the few presentation issues. I still believe, though, that more motivation and empirical results could improve the paper significantly, but I can see that the issues are not easy to address. Hence, I have chosen to improve the score I give to the paper.